# PREDICT OVERCHARGING: AUDITING LLM APIS VIA REASONING LENGTH ESTIMATION

## ABSTRACT

Commercial LLM services often conceal internal reasoning traces while still charging users for every generated token, including those from hidden intermediate steps, raising concerns of *token inflation* and potential overbilling. This gap underscores the urgent need for reliable token auditing, yet achieving it is far from straightforward: cryptographic verification (*e.g.,* hash-based signature) offers little assurance when providers control the entire execution pipeline, while user-side prediction struggles with the inherent variance of reasoning LLMs, where token usage fluctuates across domains and prompt styles. To bridge this gap, we present PALACE (*Predictive Auditing of LLM APIs via Reasoning Token Count Estimation*), a user-side framework that estimates hidden reasoning token counts from prompt–answer pairs without access to internal traces. PALACE introduces GRPO-augmented adaptation modules with a lightweight domain router, enabling dynamic calibration across diverse reasoning tasks and mitigating variance in token usage patterns. Experiments on math, coding, medical, and general reasoning benchmarks show that PALACE achieves low relative error and strong prediction accuracy, supporting both fine-grained cost auditing and inflation detection. Taken together, PALACE represents an important first step toward standardized predictive auditing, offering a practical path to greater transparency in LLM services.

## 1 INTRODUCTION

Large language model (LLM) services have made significant progress in recent years, incorporating capabilities such as multi-step reasoning (Guo et al., 2025), external tool use (Chen et al., 2024; Zhuang et al., 2023), and agent coordination (Zhao et al., 2024). These advancements enhance generation quality and expand the applicability of LLMs to a wider range of real-world domains.

However, these reasoning LLMs also produce long internal outputs that often include verbose and noisy content, such as backtracking traces or speculative reasoning branches (Aggarwal & Welleck, 2025), which may not benefit end users and can hinder comprehension. Moreover, revealing such internal traces poses security and intellectual property risks: malicious users could extract model behaviors or replicate proprietary agent workflows, threatening the commercial value of LLM systems. To mitigate these risks, LLM service providers like OpenAI (ChatGPT) (OpenAI, 2025), Google (Gemini) (Google Gemini, 2025), and Anthropic (Claude) typically withhold internal tokens generated during reasoning or tool execution. Nonetheless, users are still billed for the full sequence of tokens, including both hidden internal tokens and visible final outputs. Such services have been formally defined as *Commercial Opaque LLM Services* (COLS) (Sun et al., 2025b).

This opaque design raises notable **transparency gaps**: users are charged for token usage they cannot observe or verify. All token consumption is reported solely by the service provider, whose financial incentives may conflict with those of the user. This creates the risk of *token inflation*, where COLS may over-report token usage to increase billing. Empirical studies show that in many state-of-the-art models, over 90% of tokens are consumed in hidden reasoning, with only a small portion forming the final answer (Sun et al., 2025a). Even minor inflation under such conditions can lead to considerable increases in expenses, particularly for large-scale API users engaged in synthetic data generation, annotation, or document processing. For example, an ARC-AGI run with OpenAI's o3 model consumed 111 million tokens at a cost of $66,772 (Chollet, 2025), with over 60% from hidden reasoning tokens. Although there is no evidence of inflation in COLS, the large monetary amounts

involved mean that users have legitimate reasons to expect clearer auditing mechanisms, highlighting the need for greater transparency in LLM services.

Auditing token usage under these conditions is challenging for two reasons. First, **cryptographic approaches** (Sun et al., 2025a) require the auditor to access hidden reasoning traces in order to perform verification. In practice, however, the entire reasoning process is controlled by the COLS, making such methods impractical. Second, **user-side heuristics** based on input–output lengths are unreliable, because reasoning token usage varies drastically with domain, prompt style, and task complexity (Jin et al., 2024). Together, these challenges create a critical need for an auditing solution that works without access to hidden traces and remains robust to reasoning variability.

To address these challenges, we introduce PALACE (*Predictive Auditing of LLM APIs via Reasoning Token Count Estimation*), the first user-side framework to estimate hidden reasoning token usage directly from prompt–answer pairs. PALACE requires COLS to release lightweight auxiliary datasets that are verifiable and guaranteed to be entirely generated by COLS's own LLMs, fulfilling their obligation to provide certifiable data for auditing. Each sample in these datasets includes a user prompt, the model's reasoning process, and the final answer. PALACE addresses the **trace inaccessibility challenge** by learning to infer reasoning length from observable input–output semantics, eliminating any dependency on internal execution data. It addresses the **reasoning variability challenge** through two key innovations: a

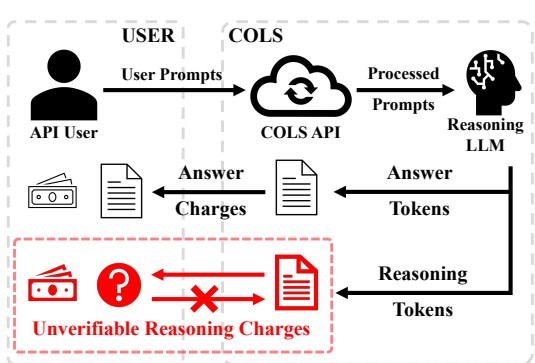

Figure 1: The transparency concern of COLS.

GRPO (Shao et al., 2024) (Group Relative Policy Optimization)-driven adaptation module that improves reasoning length inference for complex tasks, and a lightweight domain router that dynamically calibrates predictions across diverse usage scenarios.

For each auditing instance, the user provides her prompt and the corresponding response returned by COLS, along with the API configuration. PALACE invokes the router to select the appropriate GRPO adapted module, and performs the token count prediction. PALACE returns both per-sample predictions and aggregated estimates across multiple queries to support comprehensive auditing of the service. Different from prior auditing methods such as Sun et al. (2025a), PALACE neither relies on hidden tokens or intermediate embeddings during inference, nor requires users to issue additional queries or requests during the auditing process. This makes PALACE a lightweight and user-transparent auditing solution for COLS.

***Our aim is not to overstate the issue or to imply that COLS overcharge users, but to suggest that COLS offer auxiliary datasets to enable real-time auditing, strengthen trust, and enhance providers' reputations.*** Our main contributions are summarized as follows:

- We introduce reasoning length estimation as a foundation for user-side auditing of hidden reasoning token usage in commercial COLS.
- We present PALACE, which combines a GRPO-augmented adaptation module with a lightweight domain router to infer hidden reasoning length directly from prompt–answer pairs, without access to internal execution traces.
- We construct multi-domain benchmark datasets and show that PALACE achieves low relative error and stable cumulative accuracy across reasoning-heavy tasks, enabling fine-grained cost auditing and reliable detection of token inflation.

## 2 BACKGROUND AND MOTIVATION

### 2.1 COLS AND HIDDEN TOKENS

The development of reasoning LLMs (Muennighoff et al., 2025; Hao et al., 2024) and agentic LLMs (Chan et al., 2023) rapidly changes the service model of LLM APIs. As an example of a

reasoning LLM API, OpenAI now provides multiple reasoning model inference APIs such as o3, o4-mini, o4-mini-high (Jaech et al., 2024; OpenAI, 2025).

When a user submits a prompt to one of these APIs, the reasoning model typically generates a reasoning trace and the final answer. While most major providers return only the final answer in the API response, they charge based on the total number of tokens consumed by both the reasoning trace and the answer. As a form of compensation, service providers such as OpenAI, Gemini, and Claude offer users only a summarized version of the reasoning content for inspection, while the full reasoning trace remains hidden. Nevertheless, the total number of reasoning tokens, including those not visible to the user, is still accounted for in the billing. OpenAI explicitly states in its documentation: *While reasoning tokens are not visible via the API, they still occupy space in the model's context window and are billed as output tokens (OpenAI, 2025).* Similarly, Gemini notes: *When thinking is turned on, re-*

```Shell
ChatCompletion(id='chatcmpl-BfLZ5D2RW7jcCsgYCW003rLi*
****', choices=[Choice(finish_reason='stop', index=0,
logprobs=None,
message=ChatCompletionMessage(content='Let v be the
ship's <326 answer tokens, truncated for brevity> c =
(14−6)/2 = 4 km/h.', refusal=None, role='assistant',
audio=None, function_call=None, tool_calls=None,
annotations=[]))], created=1749194043,
model='o4-mini-2025-04-16', object='chat.completion',
service_tier='default', system_fingerprint=None,
usage=CompletionUsage(completion_tokens=2950,
prompt_tokens=130, total_tokens=3080,
completion_tokens_details=CompletionTokensDetails(acc
epted_prediction_tokens=0, audio_tokens=0,
reasoning_tokens=2624, rejected_prediction_tokens=0),
prompt_tokens_details=PromptTokensDetails(audio_token
s=0, cached_tokens=0)))
```

Figure 2: Example API Call to a Reasoning Model and Its Token Composition.

*sponse pricing is the sum of output tokens and thinking tokens (Google Gemini, 2025).* Specifically, Figure 2 illustrates a specific API request to a reasoning model, where the reasoning_tokens count reaches 2,624. These, together with the answer's tokens, constitute the completion_tokens that are billed at the standard output rate.

## 2.2 TOKEN AUDITING

The issue of potential price-service mismatch in COLS remains underexplored. Sun et al. (2025b) categorized such mismatches into two types: quality downgrade and quantity inflation. Quality downgrade occurs when a COLS fails to deliver the promised standard of service, instead relying on cost-saving alternatives (*e.g.,* smaller LLMs or cheaper tool calls) without informing the user. Cai et al. (2025); Yuan et al. (2025) investigated various techniques for detecting model substitution in LLM APIs, including output-based statistical tests, benchmark comparisons, log probability analysis, and the use of Trusted Execution Environments (TEEs) (NVIDIA, 2023). For quantity inflation, CoIn (Sun et al., 2025a) introduced a hash-tree-based framework that enables auditors to issue verification requests to COLS for token count and semantic validation. However, the need for hash tree generation on the server side and the auditor-COLS interaction imposes limitations on practical deployment. Thus, there is a pressing need for a high-accuracy, plug-and-play auditing method to detect quantity inflation with minimal assumptions.

## 3 DATASET CONSTRUCTION

Since prior work does not address auditing the reasoning length based solely on prompt–answer pairs, we construct the first *reasoning length auditing dataset*. Specifically, we curate four datasets comprising multi-step reasoning traces generated by DeepSeek-R1 (Guo et al., 2025) and DeepSeek-R1-Distill-Llama-70B. These datasets span diverse domains to support generalization and robust evaluation (Face, 2025):

- **OpenR1-Math**: A large-scale dataset for mathematical reasoning, consisting of 220K problems sampled from NuminaMath 1.5. Each question is annotated with 2–4 reasoning traces generated by DeepSeek-R1.
- **OpenR1-Code**: A dataset focused on program synthesis, with samples exhibiting plan-and-execute style reasoning patterns.
- **Medical**: A distilled SFT dataset generated by DeepSeek-R1 (Full Power version), targeting medically verifiable diagnostic questions based on HuatuoGPT-o1.

Table 1: Instructional prompt template for GRPO-augmented auditing model training.

---

**Prompt Template for Chain-of-Thought Reasoning with JSON Output**

```
Given a <Problem> and its corresponding <Solution>, your task is to predict how
many tokens are consumed in the process of arriving at the final <Solution> to the
problem.
Generally speaking, the more complex the problem is, the more tokens are required.

<Problem>
{problem}
</Problem>

<Solution>
{solution}
</Solution>

The Problem has {Input Length} tokens, and the Solution has {Output Length} tokens.

Please provide a detailed chain-of-thought reasoning process and include your thought
process within <think> tags.
Your final answer should be enclosed within <answer> tags.

Please return the predicted number of tokens in JSON format:
{{"count": int}}

Example format:
<think> Step-by-step reasoning, including self-reflection and corrections if
necessary. [Limited by 1024 tokens] </think>
<answer> Summary of the thought process leading to the final token count and your
predicted token count in JSON format:
{{"count": int}} [Limited by 512 tokens]
</answer>

Let me solve this step by step.
```

---

- **General Reasoning**: A large-scale synthetic dataset containing over 22 million open-domain multi-step QA examples, generated using DeepSeek-R1-Distill-Llama-70B. This dataset provides broad coverage of reasoning styles across domains.

We refer to these datasets as $\mathcal{D}_M$, $\mathcal{D}_C$, $\mathcal{D}_{\text{Med}}$, and $\mathcal{D}_G$, respectively.

**GRPO Auditing Dataset** The GRPO auditing dataset is constructed to train the auditing model under the GRPO framework. Unlike a supervised regression dataset that uses ground-truth labels for direct loss computation, this dataset provides reference values that GRPO converts into reward signals for policy optimization. Each sample consists of a structured prompt that includes a `<Problem>` section providing the user prompt, a `<Solution>` section showing the response from COLS, and explicit instructions directing the model to generate step-by-step reasoning inside `<think>` tags and output a predicted token count in JSON format within `<answer>` tags. To stabilize training and improve estimation accuracy, the prompt also provides auxiliary cues such as prompt and answer token lengths, which correlate with reasoning complexity. This structured setup, reflected in Table 1, provides the reference for reward computation, allowing GRPO to refine both the reasoning process and the accuracy of token count predictions across domains.

## 4 METHOD

### 4.1 OVERVIEW

As illustrated in Figure 3, PALACE serves as a trusted third-party auditor for commercial LLM services. It begins by acquiring a lightweight auxiliary dataset from the COLS provider and constructing an auditing dataset based on it. A general-purpose auditing model is then obtained by fine-tuning a base LLM on generic QA tasks. To accommodate diverse reasoning styles across domains and LLM APIs,

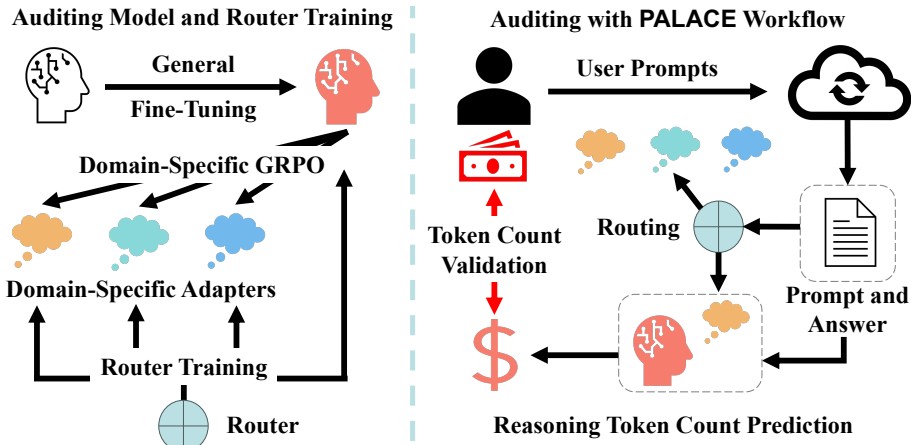

Figure 3: Workflow of PALACE.

PALACE further applies GRPO on domain-specific datasets to derive task-specific parameter deltas, referred to as GRPO-augmented modules. A lightweight router is trained to classify user prompts by domain and select the appropriate GRPO-augmented module for inference. During auditing, the user submits a prompt and the corresponding COLS output. PALACE classifies the prompt's domain, loads the relevant GRPO module, and processes the prompt–answer pair using a standardized template. The auditing model then predicts the number of hidden reasoning tokens, expanding this prediction into a tolerance interval. If the predicted token count deviates from the COLS-reported count beyond a threshold, the sample is flagged as potentially inflated. A dataset exhibiting a high proportion of such samples or a large cumulative error may indicate inflationary behavior by the COLS.

## 4.2 GRPO-AUGMENTED LLM TRAINING

To establish the auditing capability, PALACE first fine-tunes a base LLM $\mathcal{W}$ on a general QA dataset $\mathcal{D}_G$ to obtain a general-purpose auditing model $\mathcal{W}_G$. This model is trained to estimate hidden reasoning token count required for solving a given problem, based on the problem statement and the final answer:

$$\mathcal{W}_G = \arg\min_{\mathcal{W}} \ \mathbb{E}_{(x,y)\sim\mathcal{D}_G} \left[ \mathcal{L}\left(y, f_{\mathbf{W}}(x)\right)\right]. \tag{1}$$

Because reasoning complexity varies significantly across tasks and models, we further enhance $\mathcal{W}_G$ using GRPO on domain-specific datasets. For each target domain and COLS model (*e.g.,* coding domain on OpenAI's O3-mini), we perform GRPO on $\mathcal{D}_C$ to inject reasoning-specific adaptation into the model. The resulting GRPO-updated model is stored as a domain-specific delta. We define a reward function that encourages accurate predictions within a relative error threshold $\delta$ and accounts for the underlying reasoning complexity. Let $y$ be the ground-truth hidden reasoning token count and $\hat{y}$ be the known reasoning length from auxiliary data. The reward is defined as:

$$R(\hat{y}, y) = \max\left(0, 1 - \frac{|\hat{y} - y|}{|y|}\right). \tag{2}$$

Here, $\hat{y}$ denotes the model's prediction and $y$ is the ground truth target. This reward smoothly penalizes prediction error relative to the magnitude of the target value, and is clipped to lie within the $[0, 1]$ interval.

We adopt GRPO (Shao et al., 2024) as it has shown strong effectiveness on structured reasoning tasks, making it a natural fit for PALACE. RL is particularly suitable in this setting, because correctness can be directly verified to provide automatic rewards. Formally, let $\pi_\theta$ denote the policy model. For each input $Q$, the model generates $N$ candidate rollouts $\{a_{Q,1}, ..., a_{Q,N}\}$, each evaluated with explicit rewards. These per-group rollouts yield a *group-wise advantage*, which compares each rollout's

reward to the group average, reducing variance and stabilizing training. The GRPO objective is:

$$J_{\text{GRPO}}(\theta) = \mathbb{E}\left[\frac{1}{N}\sum_{i=1}^{N} \min\left(r_i A_i,\ \text{clip}(r_i,\ 1-\epsilon,\ 1+\epsilon)A_i\right)\right] - \beta D_{\text{KL}}(\pi_\theta \| \pi_{ref}), \quad (3)$$

where $r_i = \frac{\pi_\theta(a_i|Q)}{\pi_{\theta_{\text{old}}}(a_i|Q)}$ is the likelihood ratio, $A_i$ the group-wise advantage, and $D_{\text{KL}}$ a regularization term against a reference model (Ouyang et al., 2022). Integrated into the adaptation module $\Delta \mathcal{W}_C$, this objective equips PALACE with specialized reasoning-length prediction, improving accuracy on complex tasks.

## 4.3 ROUTER CONSTRUCTION

To enable domain-aware auditing, we train a lightweight router to classify user prompts into supported domains, enabling dynamic selection of the appropriate GRPO module, thereby improving prediction accuracy while avoiding the need to deploy a large monolithic model.

We denote the router as a classification function $f_{\text{router}} : x \mapsto d$, where $x$ is the user prompt and $d \in \{1, \ldots, D\}$ is the domain index corresponding to one of the GRPO-specialized domains. We train $f_{\text{router}}$ on a small subset of samples selected from each dataset $\mathcal{D}_M, \mathcal{D}_C, \mathcal{D}_{\text{Med}}, \mathcal{D}_G$, using only the prompt text and its associated domain label.

The objective is a standard cross-entropy loss over the domain labels:

$$\mathcal{L}_{\text{router}} = -\sum_{(x,d) \in \mathcal{D}_{\text{router}}} \log p(d \mid x), \quad (4)$$

where $\mathcal{D}_{\text{router}}$ is the router training set randomly sampled from the domain-specific datasets. During inference, given a new prompt $x$, the router first predicts its domain:

$$\hat{d} = f_{\text{router}}(x). \quad (5)$$

PALACE loads the corresponding GRPO module $\Delta \mathbf{W}_{\hat{d}}$, applies it to the base model $\mathbf{W_G}$, and performs reasoning length prediction with the adapted model $\mathbf{W_G} + \Delta \mathbf{W}_{\hat{d}}$. This modular design ensures scalability and efficiency while leveraging domain-specific adaptations for specific auditing tasks.

## 5 EXPERIMENTS

### 5.1 EXPERIMENTAL SETTINGS

**Models and Datasets.** We select lightweight yet capable LLMs with strong reasoning ability in experiments. For auditing models, we use Qwen2.5-1.5B (Team, 2024), Qwen2.5-3B, LLaMA3.2-1B (Dubey et al., 2024), and LLaMA3.2-3B. For benchmarks, we adopt the four datasets introduced in Section 3, covering diverse reasoning domains including mathematics, programming, medical reasoning, and general problem solving (Face, 2025). These datasets are used to evaluate PALACE and all baselines, thus providing comprehensive coverage of common reasoning tasks for large language models.

**Baselines and Metrics.** We compare our method against three baselines, including the prior work CoIn (Sun et al., 2025a) and two naive predictive auditing approaches:

- **CoIn**: A token auditing method based on an embedding hash tree and auditor-based query verification. CoIn is effective at detecting misreported token counts and meaningless token injection. However, it fails to identify repeated reasoning fragments and shows limited sensitivity when the inflation rate is low.
- **MLP**: A simple predictive auditing approach that estimates reasoning length using only the input and output lengths. We implement two two-layer MLP models, one for regression and one for classification.
- **LoRA Fine-Tuning**: A direct fine-tuning baseline where LLMs are fine-tuned on the auditing datasets with Low-Rank Adaptation (LoRA) (Hu et al., 2022) to predict the reasoning length directly.

Table 2: Prediction accuracy and error of PALACE. The upper block reports Pass@1/Pass@5 accuracy (%), and the lower block reports Average (AVG.) and Aggregated (AGG.) errors.

| Model | Method | General | | Math | | Coding | | Medical | |
|---|---|---|---|---|---|---|---|---|---|
| | | Pass@1 | Pass@5 | Pass@1 | Pass@5 | Pass@1 | Pass@5 | Pass@1 | Pass@5 |
| | **CoIn** | 63.21 | — | 44.22 | — | 38.12 | — | 46.25 | — |
| | **MLP** | 67.86 | 67.89 | 27.11 | 32.87 | 44.95 | 47.77 | 44.99 | 51.04 |
| **Qwen2.5-3B** | **LoRA** | 85.95 | 96.74 | 58.56 | 82.79 | 52.92 | 82.69 | 50.25 | 84.87 |
| | **PALACE** | **87.28** | 88.50 | **62.37** | 67.15 | **59.91** | 64.36 | **59.13** | 62.39 |
| **Qwen2.5-1.5B** | **LoRA** | 85.66 | 94.07 | 49.26 | 84.47 | 35.81 | 81.21 | 51.93 | 83.88 |
| | **PALACE** | **88.40** | 88.65 | **63.81** | 65.44 | **58.39** | 59.51 | **59.71** | 62.86 |
| **LLaMA3.2-3B** | **LoRA** | 62.74 | 89.59 | 32.53 | 59.08 | 36.40 | 53.31 | 36.99 | 60.64 |
| | **PALACE** | **81.42** | 86.86 | **51.73** | 65.60 | **65.27** | 71.15 | **59.59** | 66.68 |
| **LLaMA3.2-1B** | **LoRA** | 65.48 | 82.10 | 24.34 | 71.72 | **35.21** | 54.50 | 37.39 | 56.38 |
| | **PALACE** | **79.42** | 83.51 | **43.48** | 53.95 | 34.25 | 44.00 | **43.36** | 50.13 |
| | | AVG($\downarrow$) | AGG($\downarrow$) | AVG($\downarrow$) | AGG($\downarrow$) | AVG($\downarrow$) | AGG($\downarrow$) | AVG($\downarrow$) | AGG($\downarrow$) |
| | **MLP** | 26.39 | 1.25 | 87.45 | 2.24 | 87.44 | 1.47 | 79.02 | 1.41 |
| **Qwen2.5-3B** | **LoRA** | 18.11 | 3.93 | 40.67 | 8.74 | 40.34 | 4.07 | 39.69 | 18.24 |
| | **PALACE** | **17.82** | 6.34 | **30.58** | 6.67 | **34.76** | 5.61 | **34.09** | 13.56 |
| **Qwen2.5-1.5B** | **LoRA** | 19.03 | 5.80 | 58.63 | 2.43 | 64.61 | 1.86 | 35.82 | 33.31 |
| | **PALACE** | **18.43** | 4.80 | **36.48** | 6.72 | **48.46** | 7.75 | **33.60** | 16.68 |
| **LLaMA3.2-3B** | **LoRA** | 25.23 | 8.34 | 30.57 | 11.80 | 64.71 | 2.32 | 51.49 | 17.02 |
| | **PALACE** | **18.09** | 5.67 | **29.19** | 10.42 | **29.43** | 12.78 | **29.68** | 23.65 |
| **LLaMA3.2-1B** | **LoRA** | 30.37 | 10.20 | 58.41 | 37.19 | 52.88 | 32.11 | 50.89 | 27.61 |
| | **PALACE** | **21.65** | 16.84 | **56.98** | 15.71 | **52.74** | 38.67 | **40.62** | 55.82 |

For each dataset, we evaluate auditing performance using the following four metrics:

- **Accuracy**: A prediction is considered correct if its relative error is below 33% compared to the ground truth. Here we set the 33% threshold because, on the hardest *math* and *coding* datasets, the same LLM exhibits $\approx 33\%$ intra-model variance in hidden reasoning length for identical prompts (Sun et al., 2025b). Accordingly, if the inflation exceeds 33%, our auditing method should reliably flag it as a detectable anomaly. As shown in Figure 5, we further evaluate PALACE under thresholds ranging from 10% to 100%, where it consistently outperforms baselines.

  - **Pass@1**: Computed via greedy decoding, selecting the most likely prediction.
  - **Pass@5**: Computed via sampling five predictions with a temperature of 0.8, counting the result as correct if any prediction meets the accuracy criterion.

- **Average Error**: The mean numeric deviation between predicted and ground-truth reasoning lengths in greedy decoding.

- **Aggregated Error**: The difference between the total predicted reasoning length and the total ground-truth reasoning length, reflecting dataset-level cumulative bias.

These metrics jointly capture both the *precision* of individual predictions and the *overall bias* of the auditing model across the entire dataset. In particular, the aggregated error reflects whether small per-sample deviations accumulate into large discrepancies, which is critical for reliable large-scale auditing of hidden token usage in real-world LLM deployments.

**Training Configurations.** We use Verl (Sheng et al., 2024) as the training codebase. The relevant settings include: a generation temperature of 0.8, a total batch size of 128 (with 8 rollouts per strategy), an update batch size of 128 per GRPO step, KL penalty $\beta = 0.001$, and a learning rate of 1e-6. For the LoRA fine-tuning, we set the LoRA rank to 128, use a batch size of 128, and control the learning rate at 1e-5. Unless otherwise specified, we train for 3 epochs.

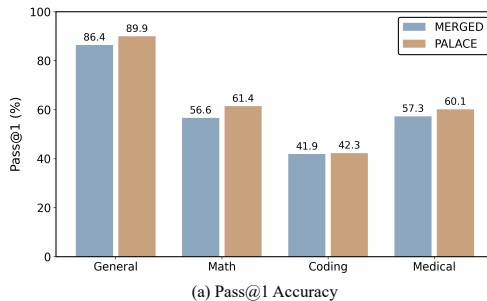 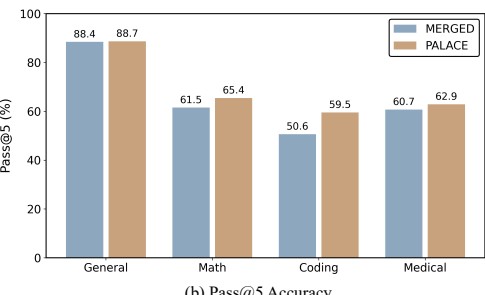

(a) Pass@1 Accuracy    (b) Pass@5 Accuracy

Figure 4: PALACE router vs LLM trained on merged dataset. Experiments are conducted on Qwen2.5-1.5B model.

## 5.2 MAIN RESULTS

**Prediction Accuracy.** Table 2 demonstrates that PALACE achieves the strongest prediction accuracy across all domains and model sizes. By leveraging semantic information rather than relying solely on text length and utilizing the reasoning abilities of the auditing model, PALACE consistently delivers high Pass@1 accuracy, even on reasoning-heavy tasks like mathematics. For instance, PALACE predicts the reasoning length with 63.81% accuracy under the 33% relative error, while baselines cannot exceed even 50% accuracy in the same setting. LoRA fine-tuning works well particularly on larger backbones such as Qwen2.5-3B. However, its performance becomes unstable on smaller models (*e.g.,* Qwen2.5-1.5B), showing signs of overfitting and limited generalization. Notably, LoRA fine-tuning has higher Pass@5 accuracy in most cases, which is due to the high uncertainty and variance of its prediction. This variance is problematic for real-world auditing, where consistent and precise predictions are essential.

For other baselines, CoIn struggles to produce reliable predictions under the 33% error boundary, failing to capture nuanced reasoning token patterns. MLP performs slightly better but is fundamentally constrained by the lack of semantic information: its predictions are shallow correlations between input–output lengths, leading to low accuracy across most domains.

**Prediction Error.** The error metrics in Table 2 further highlight PALACE's advantage. It is the only method that consistently maintains both low average and aggregated error, ensuring accurate predictions on individual samples while avoiding cumulative drift at scale. This makes PALACE uniquely suited for large-scale auditing, where aggregate consistency is critical. LoRA fine-tuning shows mixed results. Its predictions on larger models (*e.g.,* Qwen2.5-3B) appear more stable, but the improvement is limited and inconsistent across domains. On smaller models like Qwen2.5-1.5B, LoRA clearly overfits, leading to erratic error patterns and unreliable auditing performance.

The baselines are less reliable than PALACE. CoIn not only produces inaccurate predictions but also misses subtle inflation, generating high error. MLP sometimes achieves low aggregated error, which indicates it memorizes dataset-level distributions, but its high average error and low accuracy make it unsuitable for precise auditing.

**PALACE Regression vs Classification.** We further examined whether a classification LLM could serve as an alternative to PALACE's regression-based token estimation. In the classification setting, reasoning token counts are discretized into usage buckets and the model predicts the corresponding class rather than the exact number. This design allows training by simply attaching a classification head. However, as shown in Table 3, the classification approach lags behind PALACE's GRPO across

Table 3: Regression vs. classification.

| Models | Dataset | Classification | Regression |
|--------|---------|----------------|------------|
| **Qwen2.5-3B** | **General** | 62.15 | **87.28** |
| | **Math** | 56.02 | **62.37** |
| | **Coding** | 43.44 | **59.91** |
| | **Medical** | 52.10 | **59.13** |
| **Qwen2.5-1.5B** | **General** | 71.34 | **88.40** |
| | **Math** | 55.27 | **63.81** |
| | **Coding** | 51.02 | **58.39** |
| | **Medical** | 49.83 | **59.71** |

almost every dataset. On Qwen2.5-3B, classification achieves reasonable performance (*e.g.,* 62.15 on the General set), but remains notably lower than PALACE's 87.28. Similar gaps are seen in all other domains. The trend is consistent on the smaller Qwen2.5-1.5B model. These results suggest that although the accuracy of classification models is not computed in exactly the same way as regression, the data show a clear gap between classification and regression, especially GRPO based regression. A likely reason is that classification does not fully leverage the LLM's reasoning ability, which leads to weaker token count estimation. The details of classification dataset construction can be found in Appendix B.

**PALACE Router vs Merged Dataset.**    To assess whether routing truly improves auditing performance, we compared PALACE's router-based design with a simpler *merged dataset* baseline, where all domain data were combined and a single GRPO adapter was trained across tasks. As shown in Figure 4, the router consistently yields stronger results on Qwen2.5-1.5B. It achieves higher Pass@1 accuracy on nearly every subtask, from General reasoning to Medical queries, and also provides small but meaningful gains on Pass@5. This improvement comes from the router's ability to partition inputs by domain and activate the most relevant GRPO adapter, instead of forcing one generic adapter to absorb heterogeneous reasoning styles. By doing so, the router helps PALACE mitigate variance across tasks and reduces the cross-domain interference observed in the merged-training setup, resulting in more stable and accurate token-length predictions overall.

**Error Threshold and Auditing Accuracy.**    To evaluate how PALACE and baselines respond to varying error tolerances, we conduct a sensitivity analysis that adjusts the error threshold used to compute auditing accuracy. Following CoIn (Sun et al., 2025a), we gradually relax the relative error tolerance (e.g., 10%, 30%, 50%, 100%) and measure how often each method correctly flags suspicious samples under each threshold. We run this comparison on Qwen2.5-1.5B across four datasets, comparing PALACE with CoIn and LoRA fine-tuning. As shown in Figure 5, PALACE outperforms the baselines especially at low error thresholds where small deviations must still be caught. This demonstrates that PALACE is more sensitive to subtle token misreporting and better suited for real-world auditing, where providers might slightly misstate reasoning length to avoid detection.

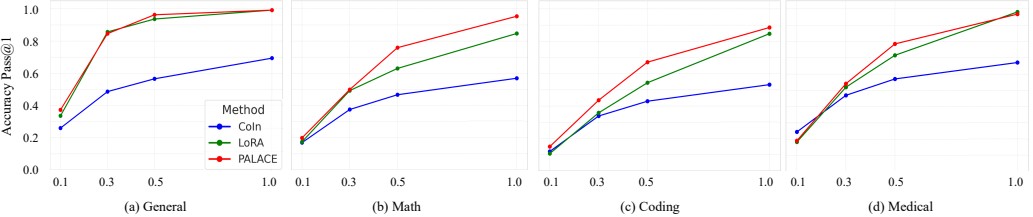

Figure 5: The relationship between error threshold (x axis) and auditing accuracy (y axis). The experiments in this figure are conducted on Qwen2.5-1.5B model.

# 6    CONCLUSION

COLS conceal most of their internal reasoning traces while still billing for every hidden token, creating a fundamental transparency gap and the risk of token inflation. In this work, we introduce PALACE, the first predictive auditing framework that estimates hidden reasoning token usage solely from prompt–answer pairs, without requiring access to internal traces. Our method combines GRPO-augmented adaptation modules and a lightweight domain router to handle reasoning style variance across domains, enabling stable predictions under heterogeneous usage scenarios. Extensive experiments on math, coding, medical, and general reasoning benchmarks demonstrate that PALACE achieves low relative error, high Pass@1 accuracy, and strong cumulative consistency. Predictive auditing is a viable path to cost verification and token inflation detection. The contributions in this paper lay a foundation for transparent and accountable LLM API billing. We hope this work draws attention to the importance of auditing LLM API billing practices and raises user awareness of the need for independent verification. By fostering such awareness, we encourage COLS to provide more accessible information for auditing, ultimately contributing to greater transparency and trust across the community.

ETHICS STATEMENT

The central goal of this research is to enhance billing transparency and accountability in LLM APIs, thereby fostering greater transparency of the commercial LLMs. Our work identifies a potential vulnerability, *token count inflation*, and proposes a prediction framework to audit the billing process. We are not suggesting, implying, or accusing any current commercial providers of engaging in such practices. Instead, we view this research as a proactive exploration of potential risks that could arise from information asymmetries between providers and users. Our objective is to contribute constructively to the ecosystem by identifying possible vulnerabilities early and proposing mitigations that help prevent the erosion of trust.

Ultimately, we believe this work contributes positively to the AI ecosystem by introducing an auditing mechanism that balances the provider's need to protect intellectual property with the user's right to verifiable billing. We therefore believe this research raises no significant ethical concerns beyond the general considerations outlined in the ICLR Code of Ethics.

REPRODUCIBILITY STATEMENT

We provide extensive details of our experimental setup in the paper for reproducing the results. The methodology for creating the training and evaluation datasets, including various prompt formats, is detailed in Section 3. Our experimental set-ups including the model and training configurations can be found in Section 5. Additional details such as the classification dataset and LoRA fine-tuning settings are further discussed in Appendix B.

Although we used LLMs to draft part of the code, all code has been manually reviewed and verified by the authors. Furthermore, algorithmic skeletons generated by LLMs were based on original Python code provided by the authors, and subsequently refined with additional details by hand. The final released codebase ensures complete reproducibility. We believe these resources provide a solid foundation for other researchers to verify our results and build upon our work.

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

## A  LLM Usage Statement

Commercial LLMs were used only as auxiliary tools and all outputs were carefully reviewed by the authors. Specifically: (i) **Preliminary zero-shot evaluation:** We tested LLMs on zero-shot prediction of reasoning token lengths to assess their auditing capability, and found that they could not reliably perform this task, which motivated the design of our training algorithm. (ii) **Code assistance:** LLMs drafted simple preprocessing scripts and converted selected Python code into LaTeX pseudocode skeletons, later verified and refined by the authors. (iii) **Manuscript polishing:** LLMs improved grammar and readability. All scientific contributions, analyses, and conclusions are solely the responsibility of the human authors. No human subjects or sensitive data were involved, and no additional ethical concerns arise beyond those stated in our Ethics Statement.

## B  Additional Experimental Setup and Results

**Hardware and Environment Setup.**  All experiments are conducted on a cluster equipped with 8 NVIDIA RTX A6000 GPUs (48GB memory each). The machines run Ubuntu 22.04 with CUDA 12.8. We use the HuggingFace Transformers library for model loading and training, and PEFT for LoRA integration. Data preprocessing and analysis are implemented in Python 3.11 with standard scientific computing libraries. This environment ensures stable large-scale training and reproducibility of our experiments.

**Classification Dataset.**  To support lightweight auditing, where a model can be trained by simply adding a classification head, we construct a classification variant of the auditing dataset. Reasoning token counts are discretized into predefined usage buckets (*e.g.,* [0–2,000], [2,000–4,000], ..., [>10,000]), each representing a cost range. This setup tolerates small deviations while still detecting major overcharges and offers a simpler alternative for models that are less suited to fine-grained regression. The dataset uses the same sources as the regression version, but each reasoning length is mapped to a bucket index. Formally, each sample is $(x, y')$, where $x$ is the same prompt–answer pair and $y' \in \{1, \ldots, K\}$ is the bucket index. In experiments, we compare this approach with GRPO. GRPO yields stronger predictions, but classification trains faster and is ideal for lightweight auditing.

In the experiments, we use five buckets (*i.e.,* five labels) for each dataset. The buckets are split so that each bucket contains roughly the same number of samples, ensuring a balanced label distribution and preventing the model from being biased toward overrepresented reasoning lengths. The bucket setting can be found in Table 4.

**LoRA Fine-Tuning Dataset.**  Since LoRA fine-tuning does not require the model to generate an explicit reasoning process, we construct a streamlined dataset where the auditing LLM is trained solely to predict the reasoning length as a regression task. This simplifies the input–output format and focuses the model's capacity on accurate numeric estimation. Table 5 illustrates the structure of the LoRA fine-tuning dataset.

Table 4: Classification bucket settings for the classification variant.

| Dataset | 0 | 1 | 2 | 3 | 4 |
|---|---|---|---|---|---|
| **General** | [0-500] | [500-700] | [700-900] | [900-1,100] | [1,100-] |
| **Math** | [0-2,000] | [2,000-4,000] | [4,000-6,000] | [6,000-10,000] | [10,000-] |
| **Coding** | [0-2,000] | [2,000-4,000] | [4,000-6,000] | [6,000-8,000] | [8,000-] |
| **Medical** | [0-500] | [500-700] | [700-900] | [900-1,200] | [1,200-] |

**Sample Efficiency of GRPO Training.**  Figure 6 shows the learning curve of the Qwen2.5-3B GRPO adapter on the Math dataset. The full Math training set contains 90K samples, but the curve plateaus roughly around the middle of training: after about 40K–50K examples, the validation Pass@1 stabilizes near its final value with only small fluctuations. This indicates that the GRPO adapter is sample-efficient and does not require using the entire dataset to reach strong performance.

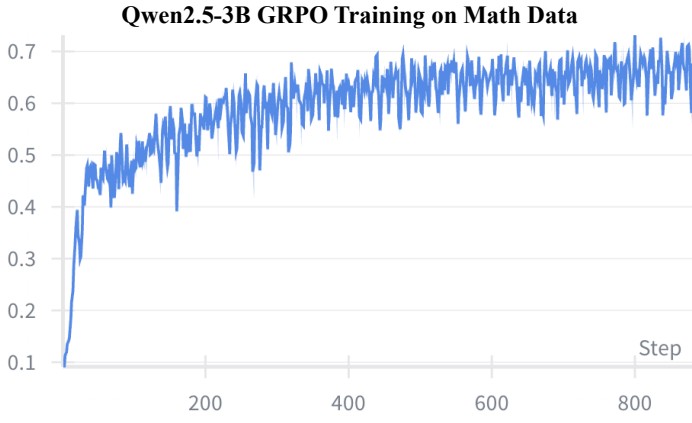

Figure 6: GRPO training curve of the Qwen2.5-3B auditor on the Math dataset (90K samples, 1 epoch). The $y$-axis shows the accuracy reward and the $x$-axis is the training step.

Table 5: Input prompt template used in regression-style reasoning length auditing.

```
Prompt Template for Regression Auditing

You are an AI reasoning analyzer.  Given a math problem and the model output
together with their token length, estimate how many tokens were used in the detailed
reasoning process that led to the answer:

Problem: {problem}

Answer: {solution}

The approximate number of tokens in the reasoning process is: {Reasoning Length}
```

## C  LIMITATIONS

While PALACE represents an initial step toward predictive auditing of hidden reasoning tokens, its current scope naturally suggests avenues for future work. The framework currently relies on a small provider-released auxiliary dataset to start training, and expanding this with more diverse or validated samples could broaden auditing coverage. Moreover, PALACE infers reasoning length only from prompt–answer pairs; incorporating complementary signals, such as lightweight metadata, may improve robustness against unusual or obfuscated reasoning traces. Taken together, these observations frame the present scope of PALACE while pointing to clear directions for making predictive auditing more flexible and widely applicable.

