# OpenReview forum: "Predict Overcharging: Auditing LLM APIs via Reasoning Length Estimation"
_ICLR.cc/2026/Conference — Submitted to ICLR 2026_

### Official Review · Reviewer_RsSN · 2025-10-26

**Soundness:** 2
**Presentation:** 2
**Contribution:** 2
**Rating:** 2
**Confidence:** 3

**Summary:**

This paper proposes to train a token count prediction model for auditing LLM APIs with hidden reasoning tokens. The goal is to detect potential overcharging by commercial opaque LLM services. The proposed method, PALACE, involves training a predictive model with supervised finetuning on pre-collected (prompt, output, token count) data and with reinforcement learning to adapt to specific domains, where the domain-specific adapters are selected with a lightweight router. The authors find PALACE outperforms several baselines and constitutes a better method for auditing LLM API token usage.

**Strengths:**

The problem they are tackling is important given the widespread usage of COLS. The methodology is simple and clear, directly addressing the task of predicting token counts with problem texts and observed outputs. The proposed method is empirically validated and outperforms baselines across benchmarks.

**Weaknesses:**

## Major weaknesses

**Lack of both theoretical, empirical, and intuitive justification of generalizability**
- The paper lacks a clear explanation of *why* reasoning length should be possible to be predicated from prompt-answer pairs. The experiments show that it is possible to fit the prediction model to specific domains, but it is unclear whether the method can generalize. It seems to me that the method would require a substantial amount of data in any new domain to be able to predict accurately on them.
- To make the method practically useful, it needs to be able to generalize OOD to unseen user prompts. For example, can a prediction model trained on code prompt-answers generalize to math and medical problems? Performing leave-one-domain-out experiments and also cross-provider evaluation can show how well the method generalizes.

**Heavy reliance on auxiliary datasets from API providers**
- The method relies heavily on provider-released public datasets, but it is unclear how large these datasets should be for the method to work on different domains. If each individual domain requires a substantial amount of data, it is unlikely that this method would be used in practice.
- To better understand this dependence, the authors can analyze accuracy vs. number of training samples across models. Intuitively, RL should require over 10K data points to work, while API providers might not be motivated enough to provide that much data for each domain.

**The proposed method is not robust to simple adversarial attacks**
- A malicious API provider could train the same model and adjust billing to stay within expected prediction bounds, while claiming token usage higher than the actual count. I suspect that the PALACE breaks down completely under this strategic provider.

**Problematic evaluation metrics and minimal improvements**
- Definition of accuracy is problematic. The paper counts a prediction "accurate" if it is within 33% of ground truth. This choice risks normalizing large billing discrepancies.
- Looking at Table 2, the AVG error differences between PALACE and the LoRA baseline on Qwen2.5-3B are often <10 tokens. Even for SOTA models like GPT-5-pro (\\$120/1M tokens), the per-query dollar difference for 10 tokens is only \\$0.0012, which seems economically tiny. This raises questions about whether additional complexity and cost (e.g., training and running the CoT predictive model and the routing model) in PALACE is necessary.

## Minor issues

Section 3
- There should be explicit definition of what data samples look like in datasets. For example, I assume they are (prompt, solution, reasoning token count) tuples? Concrete definition should be included in this section.
- Up to this section, there is no discussion on why GRPO is involved in the auditing process / training process of the auditing model. It's a bit hard to understand what the GRPO auditing dataset is used for without reading the later methodology section first.

Section 4
- Line 256: Why is ground truth different from length from auxiliary data? Also $\ell_{\text{reason}}(x)$ is defined but not used.

**Questions:**

Main questions are listed below. Some additional questions / problems are discussed in the weaknesses section.
1. What number of training samples per domain is needed to reach $\leq X \\%$ AVG error?
2. Could you include learning curves and leave‑one‑domain‑out evaluations to show whether the model generalizes?
3. Is there any theoretical justification for why it is possible to train a generalizable token number predictor?
4. Could you discuss auditor overhead brought by your method? For example, what is the prediction model’s average token usage per query under your final deployed prompt? Could you report cost ratios like (auditor tokens / suspected overcharge)?
5. If the provider adjusts reported counts using a similar prediction model, can you come up with any defense strategy?

---

> ### Author Response · Authors · 2025-11-25
> **Rebuttal: Weakness1**
>
> > W1.1: The paper lacks a clear explanation of why reasoning length should be possible to be predicated from prompt-answer pairs. The experiments show that it is possible to fit the prediction model to specific domains, but it is unclear whether the method can generalize. It seems to me that the method would require a substantial amount of data in any new domain to be able to predict accurately on them.
>
> We appreciate the reviewer’s question. Our experiments reveal an important distinction between two types of generalization. **Cross-model-family generalization** is inherently limited, because different LLM families have very different internal reasoning styles. In contrast, **generalization across reasoning tasks within the same model family** works reliably and requires only minimal calibration.
>
> Our cross-family analysis demonstrates this clearly. A predictor trained on DeepSeek-R1-Distill-Llama-70B performs very well on DeepSeek outputs, such as Qwen2.5-3B reaching Pass@1 of 87.28, but the same predictor performs poorly on a proprietary commercial model, where Pass@1 decreases to 23.40. This behavior confirms that hidden reasoning-token patterns do not transfer across model families, so a small amount of model-specific calibration is fundamentally necessary for any user-side auditing method. In practical deployment, the number of major commercial model families is very small, currently dominated by OpenAI, Google, and Anthropic, which means that an auditing-as-a-service system **only needs to maintain a few GRPO adapter groups for these providers**. Since each adapter is built on a 1.5B to 3B auditor, the overall cost remains modest and compatible with real-world feasibility.
>
> Within a single model family, PALACE generalizes much better. Our results show that we only need four to five adapters for each model family. The **General Reasoning** dataset spans a wide range of tasks, and PALACE achieves high accuracy there, often comparable to or better than specialized domains such as math or coding. This indicates that typical new tasks do not require a dedicated adapter. We introduced math, coding, and medical adapters only because many modern reasoning LLMs apply strong domain-specific optimizations to these areas, not because PALACE requires one adapter per domain. We will clarify this design choice in the revised version.
>
> We will clarify in the revision that cross-model-family generalization is not possible for any user-side estimator due to the unobservable nature of hidden reasoning traces, while cross-task generalization within families is strong and already evident in our experiments.
>
> > W1.2: To make the method practically useful, it needs to be able to generalize OOD to unseen user prompts. For example, can a prediction model trained on code prompt-answers generalize to math and medical problems? Performing leave-one-domain-out experiments and also cross-provider evaluation can show how well the method generalizes.
>
> To address the reviewer’s concern, we conducted a simple cross-domain check using the auditors already trained in our main experiments. The results are clear: the general-domain auditor transfers well to math, coding, and medical tasks, but auditors trained specifically on math, coding, or medical data do not transfer to other domains. This confirms that domain-specialized reasoning behaves differently across tasks, while the general auditor already provides strong cross-task generalization:
>
> | Train → Eval | Math | Coding | Medical | General |
> |---|----|----|----|----|
> | Math  | 63.81 | 32.14 | 9.29 | 29.27 |
> | Coding  | 24.19 |58.39| 7.31 | 18.84 |
> | Medical  | 8.36 | 9.05 |59.71| 23.55 |
> | General | **52.02** | **50.86** | **45.52** | 88.40 |
>
> Note: Each cell represents the Pass@1 accuracy of using the GRPO adapter trained on the row domain to predict the hidden reasoning length on the column domain. For example, the value in the Math column of the Coding row indicates the performance of a Coding-trained adapter when applied to Math queries.
>
> Our cross-domain analysis shows that a single general-domain auditor already transfers well across math, coding, medical, and general tasks. This indicates that PALACE does not require training a separate adapter for every new domain. We include math, coding, and medical adapters only because these domains are heavily used and modern COLS apply strong domain-specific optimizations to them. For ordinary new tasks, the general adapter already provides solid performance without any additional GRPO fine-tuning. We will add this table and a clearer explanation of this design choice in the revised version of the paper.

---

> ### Author Response · Authors · 2025-11-25
> **Rebuttal: Weakness2,3,4**
>
> > W2.1: The method relies heavily on provider-released public datasets, but it is unclear how large these datasets should be for the method to work on different domains. If each individual domain requires a substantial amount of data, it is unlikely that this method would be used in practice.
>
> Based on the results discussed in W1, PALACE does not require training a new GRPO adapter for every new domain. Our cross-domain analysis shows that a single general-domain auditor already transfers well to most tasks, and that math, coding, and medical adapters are needed only because current commercial reasoning models apply strong domain-specific optimizations in these areas. In realistic deployment, the commercial ecosystem is dominated by only a few model families, primarily OpenAI, Google, and Anthropic. Therefore, an auditing-as-a-service system would only need to maintain roughly 3×4 adapters in total for these providers: one general adapter plus three domain-specialized adapters (math, coding, medical) for each family. All adapters operate on 1.5B–3B auditors, so the resulting cost is small and entirely feasible in practice.
>
> > W2.2: To better understand this dependence, the authors can analyze accuracy vs. number of training samples across models. Intuitively, RL should require over 10K data points to work, while API providers might not be motivated enough to provide that much data for each domain.
>
> Regarding dataset size, we have included a sample-efficiency curves in the appendix on Math dataset and Qwen-3B (90K samples total, 40K samples converge). The curves show that GRPO adapters saturate quickly and do not require full datasets (math full dataset has 220K samples) to achieve strong performance. Since the final number of adapters needed for deployment is limited, the overall data requirement remains modest. We will make this connection more explicit in the revision.
>
> > W3.1: A malicious API provider could train the same model and adjust billing to stay within expected prediction bounds, while claiming token usage higher than the actual count. I suspect that the PALACE breaks down completely under this strategic provider.
>
> We appreciate the reviewer’s point. For a single query, a strategic provider could in principle adjust the reported token count to fall near the upper end of the acceptable error range. However, PALACE is not designed to rely only on per-sample accuracy. A central part of our evaluation is the aggregated error across an entire dataset, as shown in Table 2 AGG column. PALACE maintains very low cumulative error, which makes large-scale strategic inflation detectable.
>
> For users who rely on commercial APIs at scale, such as for data generation, batch processing, or agent pipelines, a provider would need to shift the reported reasoning length upward for virtually every query to create meaningful economic gain. This would cause the aggregated error to drift systematically, and PALACE would flag such inflation even if each individual query stays within a 30 percent band. In other words, a provider can hide per-sample deviations but cannot hide a consistent upward bias across hundreds or thousands of calls.
>
> We also acknowledge that no predictive auditing method can be entirely immune to adversarial providers. This is a fundamental property of any user-side estimator, and studying such adversarial strategies is an important direction for strengthening future auditing frameworks. We thank the reviewer for highlighting this scenario, and we will include a discussion of adversarial robustness in the revision.
>
> > W4.1: Definition of accuracy is problematic. The paper counts a prediction "accurate" if it is within 33% of ground truth. This choice risks normalizing large billing discrepancies.
>
> We understand the reviewer’s concern. The 33 percent threshold is not chosen to normalize large discrepancies but reflects the intrinsic variance of hidden reasoning lengths in modern reasoning LLMs. As discussed in Section 4.1 (Accuracy), the same model and the same prompt can naturally vary by roughly 30 to 33 percent. Because this level of variation comes from the model itself, a substantially lower threshold would incorrectly mark many honest traces as inaccurate.
>
> More importantly, PALACE does not rely on this threshold alone. Our auditing pipeline uses a two-level evaluation: single-sample prediction accuracy and aggregated error across large query sets. As shown in Table 2, aggregated error is consistently far smaller than 33 percent, and this metric is the primary signal for detecting systematic inflation. The per-sample threshold exists only to account for intrinsic randomness; real overcharging would appear as a persistent upward drift in aggregated error and would be easily detectable. We will clarify this design rationale in the revision.

---

> ### Author Response · Authors · 2025-11-25
> **Rebuttal: Weakness 4, Issues and Questions**
>
> > W4.2: Looking at Table 2, the AVG error differences between PALACE and the LoRA baseline on Qwen2.5-3B are often <10 tokens. Even for SOTA models like GPT-5-pro (120 dollars/1M tokens), the per-query dollar difference for 10 tokens is only 0.0012 dollar, which seems economically tiny. This raises questions about whether additional complexity and cost (e.g., training and running the CoT predictive model and the routing model) in PALACE is necessary.
>
> The interpretation here is based on a misunderstanding. The differences reported in Table 2 are **not 10 tokens, but 10 percent relative error**. Since our evaluation normalizes error by the true reasoning length, an “average error of 10” corresponds to 10 percent deviation, not ten literal tokens.
>
> The economic impact therefore depends directly on the cumulative percentage bias. Without auditing, a commercial provider could inflate token usage by 50 percent, which would cause users to pay 50 percent more for the same workload. PALACE reduces it to around 10 percent in our experiments. In practical terms, this means PALACE can eliminate approximately 40 percent of the overcharge. We will clarify this point in the revision.
>
> > I1.1: Clarifying the dataset format
>
> The GRPO dataset follows the standard structure shown in Table 1 of the paper, and includes pairs of (prompt, final answer) and other instructions. For the GRPO stage, the training samples **do not include answers inside the GRPO input itself**. Instead, GRPO uses only the prompt, and the predicted reasoning length is compared against the ground-truth reasoning-token count stored in the dataset to compute the reward.
>
> > I1.2: Why GRPO
>
> The purpose of GRPO in PALACE is to let the auditor better use semantic information from the prompt–answer pair and improve Pass@1 accuracy compared to LoRA fine-tuning. We will update the introduction to clarify this motivation earlier in the paper.
>
> > I2.1: Line 256: Why is ground truth different from length from auxiliary data? Also $\ell_{\text{reason}}(x)$ is defined but not used.
>
> The reviewer is correct to notice the inconsistency. The expression on line 256 contains a typo: the length from the auxiliary data should be $\hat{y}$ rather than $\ell_{\text{reason}}(x)$. We fix the typo in the updated manuscript.
>
> > Q1 Q2
>
> We have added experiments and discussion regarding these questions in W1.
>
> >Q3: Is there any theoretical justification for generalizable token-length prediction?
>
> Since the auditor is itself an LLM, deriving a closed-form theoretical error bound for reasoning-length prediction is not feasible. Instead, PALACE is positioned as a statistical estimator whose effectiveness comes from leveraging semantic information in the prompt–answer pair and from using GRPO to encourage deeper reasoning behavior.
>
> >Q4: The token cost of PALACE
>
> The auditor prompt used by PALACE is already shown in Table 1. Its length consists of the user prompt, the model’s final answer, and short instruction segments, and the auditor typically produces only a very small number of output tokens (the predicted token numbers).
>
> The ratio “auditor tokens / suspected overcharge” cannot be meaningfully quantified, because any such ratio would depend on the provider’s undisclosed behavior rather than on PALACE. What we can say is that PALACE uses 1.5B–3B auditors, whose compute and token cost are negligible compared to the tens- or hundreds-of-billion parameter COLS LLMs being audited.
>
> >Q5: If the provider adjusts reported counts using a similar prediction model, can you come up with any defense strategy?
>
> As discussed in W3, PALACE does not rely on single-sample accuracy alone. Auditing uses two metrics: the per-sample 33% correctness criterion and the aggregated error across queries. Even if a provider tries to mimic PALACE’s predictions, systematic inflation would still create a consistent upward drift in aggregated error, which PALACE can detect.
>
> In real datasets, some samples fall above and others below the predicted length due to inherent variance. If a provider’s reported counts are consistently above PALACE’s predictions, this pattern is itself statistically suspicious, even if each query individually stays within 33%. Such a distributional shift would be flagged easily by our aggregated-error analysis.
>
> **We sincerely thank the reviewer for the careful and detailed feedback. Your comments helped us sharpen both the motivation and the technical clarity of PALACE, especially regarding generalization, data requirements, and adversarial considerations. Since this work introduces the first predictive framework for auditing hidden reasoning tokens, our goal is to clearly articulate the problem and demonstrate that such auditing is feasible in practice. We hope the revisions and explanations provided here can convey the contribution and importance of this direction, and we would be grateful if the reviewer could reconsider the rating in light of these points.**

---

### Official Review · Reviewer_mR1Q · 2025-10-26

**Soundness:** 2
**Presentation:** 2
**Contribution:** 2
**Rating:** 4
**Confidence:** 3

**Summary:**

This paper addresses the problem of hidden reasoning tokens in commercial LLM services that are not shown to the user but still billed, raising the risk of token overcharging. The authors propose PALACE, a user-side auditing framework that estimates the number of hidden reasoning tokens from just the user’s prompt and the final answer, without access to the model’s internal chain-of-thought. PALACE works by first fine-tuning a base language model on general QA data, then adapting it to specific domains using GRPO to better predict token counts for complex reasoning tasks. It also trains a lightweight domain router that classifies each query’s domain and selects the appropriate domain-specific adapted module. During auditing, for each prompt-answer pair, PALACE predicts the hidden reasoning length and flags any large discrepancy between the prediction and the provider’s reported token usage. Experiments across four reasoning-heavy domains show that PALACE achieves low relative error in token count estimation and high accuracy under defined thresholds. This enables fine-grained cost auditing and reliable detection of any token count inflation, representing a first step toward greater transparency in LLM API billing. The key results include strong per-sample accuracy and minimal cumulative bias in predicted token totals compared to ground truth across domains.

**Strengths:**

Novel Problem: The paper identifies an important and previously under-explored transparency issue in LLM APIs and introduces the first predictive auditing framework to address it. The PALACE approach is innovative in that it requires no access to the model’s internal reasoning trace, instead leveraging a learned model to infer token usage from observable input-output semantics.

Technical Innovation: PALACE’s design is technically sound and creative. It combines reinforcement learning for fine-grained regression of token counts with domain-specific adapters and a routing mechanism. This modular approach effectively handles the high variance in reasoning lengths across different tasks and domains. The inclusion of GRPO is a strength as it allows the model to be trained with a reward that directly encourages accurate token count predictions within a relative error threshold

Strong Empirical Results: The experimental evaluation is thorough and shows clear advantages of PALACE. On math, coding, medical, and general reasoning benchmarks, PALACE consistently outperforms baseline methods in both accuracy and error measures. For instance, PALACE achieves higher Pass@1 accuracy than baselines for all tested domains and model sizes. It also maintains low average error and low aggregated error in predicted token counts, unlike simpler methods that have large per-sample errors or drift in totals.

**Weaknesses:**

Dependence on Provider Data: A practical concern is that PALACE requires auxiliary training data from the provider to calibrate the auditor for each model/domain. The framework assumes the COLS provider will release a small dataset with examples of prompts, their hidden reasoning traces, and answers. If a provider is uncooperative or if such data is unavailable, users must rely on synthetic data or open-source approximations, which may not perfectly reflect the proprietary model’s behavior. This reliance could limit PALACE’s real-world deployment unless industry standards emerge for providers to supply verifiable audit data. The authors do highlight this need and position it as fulfilling the provider’s obligation, but it remains a potential obstacle to practicality.

Computational Overhead: The PALACE auditing model is non-trivial to train and deploy. It involves fine-tuning a large language model with RL and maintaining multiple domain-specific adapter modules. The largest dataset had over 22 million synthetic examples, and GRPO training requires generating multiple rollouts per query for reward optimization. While the authors use manageable model sizes and mention using LoRA adapters to keep things lightweight, the approach still demands significant compute resources for training. This could hinder adoption by end-users or independent auditors without substantial resources. The paper could discuss more about the inference-time efficiency: running an extra LLM forward pass for each audit is necessary – which might be acceptable, but if the model is large, auditing many queries could be costly.

Generality to New Domains or Models: PALACE’s modular approach handles the four tested domains well, but it’s unclear how it generalizes to entirely new domains or to rapidly evolving model families. If a query doesn’t cleanly fall into one of the predefined domains, the router might misclassify it, affecting accuracy. The authors trained on specific domains, new reasoning domains would require additional adapter training. Similarly, adapting to a new COLS model might necessitate obtaining new calibration data and performing GRPO fine-tuning for that model. This could be time-consuming whenever providers update their models. In short, the solution might need continuous maintenance to remain effective.

Real Provider Evaluation: The experiments appear to be conducted on simulated data and open-source models where the ground-truth hidden token counts are obtainable. There is no direct test on an actual closed-source APIs, presumably because their hidden traces are not accessible. As a result, the evaluation does not demonstrate an end-to-end audit of a real commercial service’s billing. This is understandable, but it leaves a gap: the paper shows feasibility in a controlled setting, but real-world effectiveness may depend on how closely the surrogate data matches the provider’s true behavior. The authors could strengthen the work by discussing how an auditor might validate their predictions on a live API, perhaps by cross-checking cost on queries where one forces the model to reveal reasoning tokens by special API settings.

**Questions:**

To increase PALACE’s real-world impact, the authors might consider strategies to obtain or generate the necessary calibration data without full provider cooperation. For instance, could a user or auditor query a COLS with specially crafted prompts to indirectly estimate hidden token lengths? Currently, the method assumes a willing provider releasing data. Exploring semi-automated data collection or adaptation to unseen providers would make the approach more practical. Perhaps active learning could be used by querying the API and adjusting the auditor on-the-fly if the API’s behavior drifts.

---

> ### Author Response · Authors · 2025-11-25
> **Rebuttal: Weakness**
>
> > W1: Dependence on Provider Data
>
> We appreciate the reviewer’s concern. PALACE does require a small amount of model-specific calibrated data, and we agree that providers currently do not expose such information. However, this requirement is not unique to PALACE but reflects a fundamental property of the task itself. We conduct experiments on model family transfer auditing:
>
> | Model | PALACE Pass@1 (General) | Proprietary-Model Pass@1 (General) |
> |---|----|-----|
> | Qwen2.5-3B   | 87.28 | 23.40 |
> | Qwen2.5-1.5B | 88.40 | 22.88 |
> | LLaMA3.2-3B  | 81.42 | 34.01 |
> | LLaMA3.2-1B  | 79.42 | 34.95 |
>
> where GPT o-3 Pass@1 values are measured by applying the same model trained on dataset from DeepSeek-R1-Distill-Llama-70B and treating GPT o-3 reported token counts as ground truth. Experiments show this clearly: a predictor trained on Llama-70B performs well on its outputs (for example, Qwen2.5-3B has Pass@1 of 87.28) but performs poorly on GPT outputs (Pass@1 drops to 23.40).
>
> This limitation comes from the nature of the auditing problem rather than from PALACE itself. Any auditing method, including hash-based approaches, ultimately needs the provider to release at least some verified information about its reasoning behavior. Our cross-family results reinforce this point, showing that hidden reasoning-token patterns are model-specific and cannot be inferred from open-source surrogates. This is precisely why our work emphasizes the need for small, verifiable provider-released samples. PALACE requires only a very small calibration dataset, and by clearly articulating this auditing gap, we hope our work can help motivate COLS providers to release lightweight auxiliary audit data so that reliable user-side auditing becomes possible.
>
> > W2: Computational Overhead
>
> We acknowledge that training PALACE requires non-trivial compute, but this is a one-time cost for each task and does not occur during auditing. GRPO is used only to build the domain adapters; once trained, PALACE reduces auditing to a single forward pass of a 1.5B–3B model, which is already in the lightweight range if we implement PALACE to a cloud-based auditing services. Our use of small GRPO adapters significantly reduces both memory and compute cost, and the 22M examples cited refer to controlled experiments rather than the amount required in practice.
>
> In deployment, inference-time efficiency is not a concern. A 1.5B–3B auditor is orders of magnitude faster than the large reasoning models or agent-style COLS systems it is auditing, so the overhead of one extra forward pass is negligible relative to the provider’s own computation. PALACE is also intended for cloud or enterprise use, where inference remains inexpensive. Our experiments show that 1.5B and 3B auditors already saturate accuracy, so larger models are unnecessary. We will clarify the difference between training cost and auditing cost in the revision.
>
> > W3: Generality to New Domains or Models
>
> We acknowledge that adapting PALACE to a new domain or a new COLS model requires minimal calibration, but this is a property of the hidden-reasoning auditing task itself rather than a limitation of PALACE. Our cross-family results in W1 show that reasoning-token behavior does not transfer across model families, which means that any effective auditing method must incorporate at least a small amount of model-specific data. PALACE addresses this requirement in the most lightweight way possible by using small GRPO adapters rather than retraining the full auditor, so extending to a new domain or model by such a plug-in-and-play method only requires limited additional training. Besides, our experiments on General Reasoning dataset shows the auditing performance across multiple tasks (often has better overall performance than specific tasks such as math and coding).
>
> In practice, the number of commercially deployed reasoning models is small and relatively stable, and reasoning-heavy domains such as math, coding, medical, and general QA cover the majority of real-world API usage. This means that the maintenance burden is modest. PALACE’s contribution is to clearly articulate this unavoidable calibration requirement for reliable auditing and to provide an efficient architecture that minimizes adaptation cost. We will clarify this point in the revision.

---

> ### Author Response · Authors · 2025-11-25
> **Rebuttal: Weakness and Questions**
>
> > W4: Real Provider Evaluation
>
> We thank the reviewer for raising this important point and for acknowledging that the limitation is inherent to the current ecosystem. Since no commercial COLS exposes verified hidden reasoning traces, it is not yet possible for any auditing method to conduct a full end-to-end evaluation on a closed-source API. This is precisely why we rely on open-source models and high-quality reasoning datasets where ground-truth traces are available at scale. We appreciate the reviewer’s understanding, and agree that this situation further reinforces our motivation: if hidden-token auditing is important and the risk of inflation is real, then the current lack of provider-released audit data represents a significant gap in transparency for users. We are grateful that the reviewer recognizes the value of highlighting this problem.
>
> Regarding how an auditor might validate its predictions on a live API, we appreciate this insightful suggestion. Several feasible strategies exist even without access to internal traces. One approach is to evaluate PALACE on queries where the provider voluntarily exposes partial reasoning signals, such as constrained outputs, structured chain-of-thought summaries, or cost-controlled prompts. Another approach is to use consistency checks across difficulty-controlled prompts, allowing the auditor to estimate whether the reported reasoning-length trends align with semantic complexity. A third possibility is active learning, where the auditor periodically probes the API with diagnostic queries to detect drift or unexpected changes in token accounting. We will include a discussion of these potential pathways in the revision, as they represent promising directions for validating auditing methods on live commercial systems.
>
> > Q1: To increase PALACE’s real-world impact, the authors might consider strategies to obtain or generate the necessary calibration data without full provider cooperation. For instance, could a user or auditor query a COLS with specially crafted prompts to indirectly estimate hidden token lengths? Currently, the method assumes a willing provider releasing data. Exploring semi-automated data collection or adaptation to unseen providers would make the approach more practical. Perhaps active learning could be used by querying the API and adjusting the auditor on-the-fly if the API’s behavior drifts.
>
> We appreciate the reviewer’s suggestion and agree that developing strategies to obtain calibration data without explicit provider cooperation would greatly increase the practical impact of PALACE. Since this work is an early exploratory study, our primary goals are to clearly articulate the auditing problem and demonstrate that predictive estimation is feasible once minimal ground truth exists.
>
> The idea of crafting prompts that elicit indirect signals about hidden reasoning length is very interesting. Conceptually, it resembles a form of “reasoning-length jailbreaking,” where the auditor tries to induce the COLS model to reveal usable information without exposing the full chain-of-thought. We find this direction promising because it does not rely on a fully cooperative provider and could serve as a lightweight alternative source of calibration data.
>
> At the same time, such probing-based methods may face challenges in scalability and may risk subtly affecting model outputs, but they offer an elegant pathway toward semi-automated adaptation and drift detection. We appreciate this insightful suggestion and will incorporate a discussion of these possibilities in the revision, as they represent valuable future directions beyond the scope of the present foundational work. We will also include this into our revised paper when talking about potential future works.
>
> -------------------------------------------
>
> **We would like to sincerely thank the reviewer for the unusually thoughtful and insightful feedback. We deeply appreciate that the reviewer fully understood the real-world scenario behind our work and recognized the importance of the auditing problem we raise. We are also grateful for the reviewer’s creative suggestions on improving PALACE, including strategies for collecting calibration data without provider cooperation. These ideas are genuinely valuable to us and will meaningfully influence our future research. In this rebuttal, we have addressed each concern carefully, and we hope the reviewer can take into account that this paper introduces the first predictive auditing framework, demonstrates its feasibility through systematic experiments, and highlights a problem that directly impacts every user in the age of large-scale reasoning models. We would be grateful if the reviewer could reconsider the score in light of these contributions and the importance of the direction.**

---

### Official Review · Reviewer_6xRC · 2025-10-29

**Soundness:** 2
**Presentation:** 2
**Contribution:** 2
**Rating:** 4
**Confidence:** 3

**Summary:**

Although commercial opaque LLM Services do not show internal reasoning traces, users still pay for every generated token, which raises concerns of token inflation and potential over-billing. In light of this, the authors introduce PALACE (Predictive Auditing of LLM APIs via Reasoning Token Count Estimation), a user-side framework that estimates hidden reasoning token counts from prompt–answer pairs without access to internal traces. They demonstrates the efficacy of PALACE in multiple domains such as math, coding, medical, and general reasoning benchmarks.

**Strengths:**

1. Estimating hidden reasoning token counts from prompt–answer pairs without access to internal traces seems an interesting and new idea.
2. PALACE shows the potential for an important first step toward standardized predictive auditing.
3. It is well-written and easy-to-follow.

**Weaknesses:**

1. It is unclear why PALACE can work well. Can the authors provide more detailed analysis (e.g., a theoretical analysis)?
2. All the experiments are done with 1B or 3B-sized LLMs. It would be more beneficial if the authors conduct experiments for 8B-sized LLMs such as Llama 3.1 8B or Qwen3 8B.
3. In Table 2, PALACE usually outperforms LoRA for Pass@1, but underperforms for Pass@5. It is hard to understand why such a phenomenon happens. Can the authors explain this in detail?

**Questions:**

1. When reporting Pass@5, the authors used a temperature of 0.8, but the recommended temperature is different depending on the chosen model. Is there any reason why the authors used the same temperature for different models?

---

> ### Author Response · Authors · 2025-11-25
> **Rebuttal: Weakness**
>
> > W1: It is unclear why PALACE can work well. Can the authors provide more detailed analysis (e.g., a theoretical analysis)?
>
> We agree that providing a full theoretical guarantee is fundamentally difficult in this setting, since hidden reasoning traces are inaccessible and reasoning-token behavior cannot be bounded analytically. Instead, PALACE is supported by empirical evidence and by the structure of the predictive task itself. The accuracy of reasoning-length estimation depends on three key factors:
>
> (1) the underlying model family and its reasoning style,
>
> (2) the semantic content of the prompt–answer pair, and
>
> (3) the capacity and training objective of the auditing model.
>
> Compared to baselines, PALACE improves all three aspects.
>
> First, naive regression/classification baselines use only token lengths as input and cannot extract semantic cues from the prompt or answer, which limits their ability to model factor (2). Second, hash-based approaches do not use LLMs as auditors, and therefore cannot leverage semantic structure either. Third, compared to LoRA-based fine-tuning, PALACE benefits from GRPO training, which injects reasoning capability into the auditor and substantially improves factor (3). Meanwhile, we keep factor (1) controlled and provide lightweight GRPO adapters so that PALACE aligns with each model family without retraining from scratch.
>
> Because PALACE jointly captures semantic information, model-specific reasoning patterns, and stronger auditor training, it achieves the best overall performance among all baselines. We will clarify this intuition in the revision.
>
> > W2: All the experiments are done with 1B or 3B-sized LLMs. It would be more beneficial if the authors conduct experiments for 8B-sized LLMs such as Llama 3.1 8B or Qwen3 8B.
>
> We appreciate the reviewer’s suggestion to include larger auditor models. In practice, GRPO training is computationally expensive, and running GRPO on an 8B backbone is far beyond our system capacity: on our 8 A6000 GPUs, training an 8B auditor with GRPO would require over 500 hrs. This is not feasible within resource limits, and we hope the reviewer can understand this constraint.
>
> From an application perspective, PALACE is intended to serve as a real-time or large-scale auditing component, which is also noted by Reviewer e5tK. Larger auditors such as 8B–14B models may make auditing services impractical due to latency and cost constraints. Our experiments show that 1.5B and 3B models already achieve similar performance, suggesting that scaling to larger models may offer diminishing returns.
>
> For these reasons, we chose 1.5B and 3B auditors as a practical balance: they provide high accuracy while remaining computationally viable for both our experiments and for realistic deployment scenarios.
>
> > W3: In Table 2, PALACE usually outperforms LoRA for Pass@1, but underperforms for Pass@5. Can the authors explain this in detail?
>
> We thank the reviewer for raising this point. This behavior is expected given the difference between GRPO and supervised LoRA fine-tuning. GRPO is an RL-based objective that explicitly upweights high-reward outputs and downweights all others, which in our setting encourages the auditor to produce predictions that fall within the 33% relative-error range. This mode-seeking behavior naturally improves Pass@1 accuracy but **reduces output diversity**. Prior work on policy-gradient finetuning has shown that RL objectives typically collapse the output distribution toward high-reward modes and reduce entropy[1,2].
>
> In contrast, LoRA fine-tuning optimizes a log-likelihood objective, which preserves more output diversity. As a result, LoRA sometimes generates a few “near-miss” predictions within the acceptable error range, improving Pass@5, although its top-1 prediction is less stable.
>
> Because auditing ultimately requires accurate single predictions rather than diverse ones, Pass@1 is the more meaningful metric for this task, and this is precisely why we use GRPO training.
>
> > Q1: When reporting Pass@5, the authors used a temperature of 0.8, but the recommended temperature is different depending on the chosen model. Is there any reason why the authors used the same temperature for different models?
>
> We use a fixed temperature of 0.8 to ensure fair comparison across different auditors; model-specific “recommended” temperatures are tuned for generation quality, not for evaluating Pass@5 diversity. Pass@1 and error metrics use greedy decoding and are unaffected by temperature.
>
> Reference
>
> [1] Stiennon, N., Ouyang, L., Wu, J., Ziegler, D., Lowe, R., Voss, C., ... & Christiano, P. F. (2020). Learning to summarize with human feedback. Advances in neural information processing systems, 33, 3008-3021.
>
> [2] Ouyang, L., Wu, J., Jiang, X., Almeida, D., Wainwright, C., Mishkin, P., ... & Lowe, R. (2022). Training language models to follow instructions with human feedback. Advances in neural information processing systems, 35, 27730-27744.

---

### Official Review · Reviewer_e5tK · 2025-10-31

**Soundness:** 2
**Presentation:** 3
**Contribution:** 2
**Rating:** 2
**Confidence:** 3

**Summary:**

Nowadays, many commercial opaque LLM services (COLS) need hidden reasoning states which do not reveal to users but could cause additional charge. The authors argue that COLS can lead to potential overcharging, and a user-side auditing framework estimating hidden token usage is necessary. They introduce a GRPO-augmented model combined with a lightweight domain router to predict the number of hidden reasoning tokens. Experiments on several synthetic benchmarks (math, coding, medical, and general reasoning) demonstrate that PALACE achieves lower relative error and higher accuracy compared to baselines such as CoIn, LoRA fine-tuning, and MLP regression. The paper positions predictive reasoning-length estimation as a practical step toward transparent LLM billing.

**Strengths:**

[S1] Addresses an important and underexplored issue: auditing and transparency of commercial LLM billing.

[S2] The paper is clearly and well written.

**Weaknesses:**

[W1] Lack of generalization. All datasets and reasoning traces come from the DeepSeek family, with no evidence that PALACE generalizes to other LLM families (e.g., OpenAI, Anthropic, Gemini). Without such validation, it is unclear whether PALACE can audit COLS in practice.

[W2] No cross-model or out-of-domain (OOD) robustness evaluation. The paper does not study whether PALACE trained on one reasoning model can handle prompts or outputs from a different one. If not, each new COLS model, or even each model update, would require retraining from scratch, which severely limits practical deployability.

[W3] Evaluation methodology is weak. The 33% relative error threshold for correctness is arbitrary and not rigorous. Metrics such as Average and Aggregated Error are only reported for greedy decoding, yet the results appear sensitive to sampling temperature. A more systematic evaluation of variance and confidence calibration is needed.

[W4] Questionable real-world feasibility. Although PALACE is described as a lightweight and user-side framework, it relies on auditor models with 1.5B–3B parameters (e.g., Qwen2.5, LLaMA3.2), which entail substantial computational and memory demands. This raises concerns about the practicality of deploying PALACE for real-time or large-scale auditing, particularly for individual or enterprise users without significant resources. Furthermore, the approach assumes COLS will provide verified reasoning traces to bootstrap PALACE training. This assumption is unrealistic in most commercial contexts, and without such data the framework cannot be initialized.

**Questions:**

- Could PALACE show high generalization to other model families? It would be better to provide evidence that PALACE can work on other model families and COLS. (see W1)

- Does PALACE have cross-model and OOD robustness? (see W2) If PALACE does not show high cross-model and OOD robustness, the system would require auxiliary datasets every time the model changes. Connected to W4, without cross-model and OOD robustness, it would have very low feasibility in real-world scenarios.

-  Why did you choose the 33% relative error threshold? How do the average and aggregated errors change when the sampling temperature varies? Comparing average and aggregated errors under non-greedy decoding would be more informative than comparing only Pass@1. (see W3)

---

> ### Author Response · Authors · 2025-11-24
> **Rebuttal: Weakness**
>
> We thank the reviewer for engaging deeply with the submission. Below we address each concern. Since the reviewer raises fundamental questions about feasibility and generalization, we respond with additional clarity on the motivation and scope of PALACE.
>
> >W1: Lack of generalization. All datasets and reasoning traces come from the DeepSeek family, with no evidence that PALACE generalizes to other LLM families (e.g., OpenAI, Anthropic, Gemini).
>
> We appreciate the reviewer’s concern regarding generalization beyond the DeepSeek family. We conducted a simple cross-family test to better understand this issue. We used a PALACE predictor trained on DeepSeek-R1-Distill-Llama-70B traces to estimate hidden reasoning-token usage for ChatGPT outputs (treating the provider-reported counts as ground truth). As shown below, the predictor’s Pass@1 accuracy drops sharply when transferred to GPT o-3, confirming that hidden reasoning-token behavior differs substantially across model families:
>
> | Model         | PALACE Pass@1 (General) | ChatGPT Pass@1 (General) |
> |--------------|--------------------------|---------------------------|
> | Qwen2.5-3B   | 87.28                    | 23.40                     |
> | Qwen2.5-1.5B | 88.40                    | 22.88                     |
> | LLaMA3.2-3B  | 81.42                    | 34.01                     |
> | LLaMA3.2-1B  | 79.42                    | 34.95                     |
>
> These results indicate that cross-family reasoning-length prediction does not transfer well in practice, as each LLM family exhibits its own reasoning style. However, this is not a limitation specific to PALACE, but a property of the underlying auditing problem: **without minimal model-specific auxiliary data, no predictive method can reliably audit hidden reasoning tokens of a COLS**.
>
> Therefore, rather than weakening PALACE’s contribution, this observation reinforces our core motivation: current COLS designs make auditing impossible because users have no access to even minimal verified samples. PALACE clarifies this requirement and provides the first concrete framework showing that once a provider releases a small, verifiable auxiliary dataset, user-side auditing becomes feasible and accurate. We will make this motivation clearer in the revision.
>
> >W2: No cross-model or out-of-domain (OOD) robustness evaluation.
>
> For cross-model and OOD robustness, we refer the reviewer to the table in W1: a predictor trained on Llama model performs poorly when applied to ChatGPT outputs, showing that hidden reasoning-token behavior is intrinsically model-specific and does not support reliable OOD transfer for any predictive method.
>
> **However, this does not mean PALACE must be retrained from scratch for every new COLS model.** As described in the paper, PALACE trains one general auditor and adds lightweight GRPO LoRA adapters for each model family. These adapters are small, fast to train, and require only limited auxiliary data. Given that commercially deployed reasoning models are few (primarily OpenAI, Anthropic, Google), an auditing-as-a-service system would only need a small number of such adapters. We will clarify this in the revision.
>
> >W3: Evaluation methodology is weak. The 33% relative error threshold for correctness is arbitrary and not rigorous.
>
> The 33% relative-error threshold is **not arbitrary**. As we claimed in **Section 5.1(Accuracy)**: it directly reflects empirical variance observed in the original dataset we used. As reported in CoIn (Sun et al. (2025b)), the same reasoning model can exhibit ≈33% intra-model variability in hidden reasoning length for the exact same prompt due to natural stochasticity in chain-of-thought generation. Our dataset displays the same phenomenon: identical prompts from DeepSeek-R1 traces vary in length by up to ~30% because of backtracking, self-corrections, and alternative branches. The 33% threshold therefore corresponds to the intrinsic noise floor of real reasoning LLMs, not a design choice made for convenience.
>
> The reasoning traces in our benchmarks are naturally collected, already containing sampling randomness and multi-path reasoning variability. PALACE’s performance under this realistic stochasticity remains consistently strong, while baselines degrade much more. Because our evaluation uses real traces rather than artificially controlled decoding, the reported results already capture the variation the reviewer is concerned about. We will make this connection clearer in the revision.

---

> ### Author Response · Authors · 2025-11-24
> **Rebuttal: Weakness and Questions**
>
> > W4: Questionable real-world feasibility. Although PALACE is described as a lightweight and user-side framework, it relies on auditor models with 1.5B–3B parameters (e.g., Qwen2.5, LLaMA3.2), which entail substantial computational and memory demands. This raises concerns about the practicality of deploying PALACE for real-time or large-scale auditing, particularly for individual or enterprise users without significant resources. Furthermore, the approach assumes COLS will provide verified reasoning traces to bootstrap PALACE training. This assumption is unrealistic in most commercial contexts, and without such data the framework cannot be initialized.
>
> We clarify that “user-side auditing” does not mean running PALACE on edge devices. In realistic deployments, auditing would be performed as a cloud or enterprise service, where models in the 1.5B–3B range are already considered lightweight (reviewer 6xRC even noted that using a larger 8B backbone could further improve accuracy). These sizes fit comfortably on a single modern GPU, making them practical for batch or periodic auditing rather than per-query realtime use.
>
> Regarding initialization data, PALACE does rely on a small amount of model-specific auxiliary traces. This is not a limitation unique to PALACE but a fundamental requirement of hidden-token auditing: as our cross-model results show, reasoning-length behavior is inherently model-specific and cannot be transferred across LLM families. Therefore, any practical auditing framework requires minimal verified samples. Our contribution is to clearly articulate this requirement, demonstrate that predictive auditing becomes feasible once such data exist, and highlight why standardized auxiliary auditing datasets are necessary for transparency in COLS.
>
> > Q1: Generalization to other model families
>
> PALACE does not directly generalize across LLM families, as shown by the reduced performance when a Llama/Deepseek-trained predictor is applied to outputs from a different proprietary model family. This reflects the intrinsic model-specific nature of hidden reasoning tokens and the need for minimal model-specific auxiliary data.
>
> > Q2: Cross-model and OOD robustness
>
> Hidden reasoning-token behavior does not transfer across LLM families, so cross-model and OOD robustness are inherently limited. PALACE addresses this realistically by using one general auditor with lightweight GRPO adapters, and only a small number of such adapters are needed for today’s few major commercial reasoning models.
>
> > Q3: Choice of 33% threshold and temperature variance
>
> The 33% threshold follows empirical evidence in Section 5.1 and prior work showing that identical prompts naturally vary by about 30–33 percent in hidden reasoning length. Our benchmarks already contain this natural sampling variability, and PALACE remains stable under these realistic conditions while baseline methods degrade.\
>
> **We fully recognize that hidden reasoning-token auditing is fundamentally challenging, and the current ecosystem offers no cross-model or OOD path because reasoning behavior itself is model-specific. This is exactly why we position PALACE as the first predictive auditing framework: our goal is to clearly surface this under-explored but important problem, articulate the minimal conditions under which auditing becomes possible, and provide a concrete solution that works well once these conditions are met. The lightweight adapter design reflects our honest acknowledgment of the limitation and our intent to make model-specific adaptation as practical as possible. We hope the reviewer can see that our main contribution lies in raising an important auditing question, proposing the first predictive approach, and demonstrating empirically that it is effective and substantially stronger than prior methods. We would greatly appreciate the reviewer’s reconsideration of the score in light of these clarified motivations and contributions.**

---

> > ### Comment · Reviewer_e5tK · 2025-11-26
> >
> > Thank you for your responses.
> >
> > Regarding the motivation of the predictive method, the author's explanation partially addresses my concerns about generalization and OOD robustness. While achieving full cross-model and cross-dataset generalization would indeed be ideal, and I understand that it is difficult in practice.
> >
> > However, my concern about the evaluation methodology remains unresolved. The justification for the 33% relative error threshold is still not convincing. A 33% margin is considerable, especially when this threshold is primarily motivated by one prior work. If authors claim that 33% variation of reasoning lengths is natural, this may weaken the motivation of this work, as it suggests that hidden reasoning lengths fluctuate so widely that obtaining precise or reliable estimates becomes impossible.
> >
> > In addition, the issue of sampling variance (e.g., temperature changes) has not been sufficiently addressed. The paper does not report Average or Aggregated Error across different decoding temperatures, and while Fig. 5 includes results at a smaller threshold (10%), performance declines noticeably (e.g., below 0.4). This indicates that the method may be sensitive when evaluated under stricter prediction criteria.
> >
> > In my view, these points raise important considerations that would benefit from further clarification.

---

> > > ### Author Response · Authors · 2025-11-26
> > > **Response to reviewer's comments**
> > >
> > > We thank the reviewer for the thoughtful follow-up and for raising important points. We respond as follows.
> > >
> > > > However, my concern about the evaluation methodology remains unresolved. The justification for the 33% relative error threshold is still not convincing. A 33% margin is considerable, especially when this threshold is primarily motivated by one prior work. If authors claim that 33% variation of reasoning lengths is natural, this may weaken the motivation of this work, as it suggests that hidden reasoning lengths fluctuate so widely that obtaining precise or reliable estimates becomes impossible.
> > >
> > > We agree that a 33% per-sample margin reflects the substantial natural variation in modern reasoning models. However, we believe this does not weaken the motivation of our work. Instead, it highlights why auditing is necessary. The challenge we address is that hidden reasoning lengths fluctuate widely and users currently have no way to detect systematic inflation. **This variance of reasoning traces is therefore not a limitation of PALACE but an inherent difficulty of the predictive auditing task itself.**
> > >
> > > To address this issue, PALACE does not rely on per-sample accuracy alone. **We jointly evaluate inflation risk using both the proportion of samples within 33% error range and the aggregated error across all queries.** As shown in Table 2, cumulative error remains low in practice, often within 10 percent on Qwen2.5-3B, which reflects the overall consistency of the auditor when API calls are considered at scale.
> > >
> > > In practical auditing, PALACE relies on these two indicators together. When both indicators become unusually high, for example when more than 40% samples exceed the error range and the aggregated deviation grows beyond 20%, the likelihood of inflation becomes significant. By jointly using these two metrics, PALACE can remain reliable even under the inherent variability of reasoning traces and can effectively detect systematic over-reporting.
> > >
> > > >In addition, the issue of sampling variance (e.g., temperature changes) has not been sufficiently addressed. The paper does not report Average or Aggregated Error across different decoding temperatures, and while Fig. 5 includes results at a smaller threshold (10%), performance declines noticeably (e.g., below 0.4). This indicates that the method may be sensitive when evaluated under stricter prediction criteria.
> > >
> > > We agree that our experiments do not include datasets generated at multiple explicit temperatures. This is because our auditing uses the Open-R1 distillation data, which does not record sampling configurations. Community discussions indicate that the reproduction commonly uses a medium-to-high temperature around 0.6, and this matches what we observe in the data: even identical prompts show roughly 30% variation in reasoning length. This variance therefore comes from the dataset itself rather than our choice of evaluation.
> > >
> > > Since both training and evaluation occur under this setting, PALACE’s performance reflects the realistic variance of the data. In such a medium-to-high-variance setting, PALACE remains far more accurate than non-predictive baselines (Fig. 5), and it is understandable that Pass@1 drops when the tolerance is tightened to ten percent, because even two generations of the same prompt frequently differ by more than this amount.
> > >
> > > If data were generated at lower temperature, the natural variance and the appropriate error tolerance would also be smaller. Studying how decoding settings influence the variance of reasoning length is an interesting direction for future work, but collecting multi-temperature reasoning traces is beyond our current scope. Under the medium-variance regime available to us, PALACE achieves state-of-the-art accuracy.

---

### Official Review · Reviewer_wSQr · 2025-11-04

**Soundness:** 3
**Presentation:** 3
**Contribution:** 3
**Rating:** 6
**Confidence:** 2

**Summary:**

The paper introduces PALACE, a framework for auditing commercial reasoning LLM APIs by predicting hidden reasoning token usage from visible prompt–answer pairs. Using GRPO-based modules and a router across domains, PALACE estimates token counts without internal access to provider traces and can detect of overcharging in LLM API services. Experiments on across datasets from different domains show improved accuracy and low error compared to multiple baselines.

**Strengths:**

1. The paper addresses an increasingly relevant problem of auditing hidden reasoning token usage in LLM APIs. It also proposes a new evaluation framework in addition to existing work of CoIn.

2. Innovatively combine a domain router which improving prediction accuracy across different reasoning domains.

3. The experimental setup is systematic, including both regression and classification formulations and clear error analyses.

**Weaknesses:**

1. In real commercial API services, inference-time randomness and determinism settings can vary substantially across providers. How would PALACE perform under such conditions, and can you simulate varied inference environments to demonstrate the robustness of its auditing estimates?

2. The method’s reliance on calibrated reference data implies partial access to provider-side information, which reduces its practicality as an user-side auditing framework.

3. The comparison to CoIn remains mostly empirical; there is little discussion of theoretical guarantees or conditions where PALACE might fail.

**Questions:**

1. Could the model-based estimation in PALACE be integrated with CoIn’s verification to combine predictive power with cryptographic reliability?

2. Have you tested whether PALACE’s GRPO-based length predictor generalizes across model families trained with different reasoning styles?

---

> ### Author Response · Authors · 2025-11-24
> **Rebuttal: Weakness**
>
> We thank the reviewer for the thoughtful comments and positive assessment of the paper’s motivation, methodology, and experimental design. Below we provide concise responses to the weakness and questions.
>
> >W1: In real commercial API services, inference-time randomness and determinism settings can vary substantially across providers. How would PALACE perform under such conditions, and can you simulate varied inference environments to demonstrate the robustness of its auditing estimates?
>
> PALACE does not rely on deterministic reasoning traces.
> Importantly, the datasets used in our experiments (e.g., OpenR1-Math, Coding, Medical, General) are naturally collected from real reasoning LLMs such as DeepSeek-R1 and inherently contain substantial randomness such as backtracking, self-corrections, and multi-branch variability. Our experiment (Section 5.1, Accuracy) shows that on math and coding tasks, the **same model** and **same prompt** in our dataset can exhibit ≈33% intra-model variance in hidden reasoning length. This randomness is therefore already present in our evaluation setting.
>
> Under such realistic variance, PALACE consistently achieves strong Pass@1 accuracy and low aggregated error, as shown in Table 2 and Figure 5. In contrast, baselines that rely more directly on stable token patterns (e.g., CoIn) degrade sharply under this natural variance and require a much higher detection threshold to remain effective, demonstrating PALACE’s superior robustness.
>
> We will clarify this point to emphasize that PALACE is evaluated under realistic stochastic reasoning behavior, not deterministic traces.
>
> >W2: The method’s reliance on calibrated reference data implies partial access to provider-side information, which reduces its practicality as an user-side auditing framework.
>
> We appreciate the reviewer’s observation and agree that major commercial COLS (e.g., OpenAI, Google, Anthropic) currently do not provide certified reasoning-trace datasets for auditing, which naturally limits transparency. **This situation directly confirms the real-world relevance of the problem we raise: users today lack any verifiable mechanism to audit hidden reasoning-token billing.**
> Thus, the need for a lightweight provider-released reference dataset is not a limitation of PALACE itself, but a reflection of the existing absence of standardized auditing support in commercial LLM services (not implying intentional inflation, but highlighting that the system design makes verification impossible).
>
> Our experiments further indicate that effective auditing requires reference data from the same model family, rather than relying on cross-model transfer. Different LLMs exhibit substantially different reasoning-length distributions and stylistic preferences, making transfer-only auditing unreliable. This empirical finding reinforces the need for minimal, model-specific auxiliary samples: without them, no user-side method can accurately estimate hidden reasoning tokens under realistic variance.
>
> From this perspective, the requirement for lightweight reference data is not an assumption of convenience, but an inevitable condition for any practical auditing framework. PALACE simply makes this requirement explicit and demonstrates that, once such minimal data are provided, user-side auditing becomes feasible and effective.
>
> >W3: The comparison to CoIn remains mostly empirical; there is little discussion of theoretical guarantees or conditions where PALACE might fail.
>
> We agree with the reviewer that theoretical guarantees’ are fundamentally limited in our setting. Since COLS do not expose internal reasoning traces, any user-side estimator must be statistical: by definition, it cannot be proven correct for every query without access to the underlying trace.
>
> For this reason, PALACE is explicitly framed as a predictive auditing framework based on reasoning models. Its guarantees are therefore statistical: given model-specific auxiliary data, PALACE minimizes relative error (Eq. 2) and achieves low aggregated error over large query sets, so that any persistent inflation produces a systematic drift that becomes detectable as more samples accumulate. What we can rigorously characterize is the behavior of this estimator in expectation and at scale, not worst-case correctness for every individual call.
>
> We will make this limitation explicit and also clarify conditions under which PALACE may fail in practice, such as severe distribution shift between auxiliary data and deployed usage, or intentionally obfuscated reasoning styles that break the semantic link between prompt–answer pairs. These constraints are inherent to all user-side predictive approaches, and our contribution is to show that, under realistic assumptions and naturally collected reasoning data, such predictive auditing can still be practically useful.

---

> ### Author Response · Authors · 2025-11-24
> **Rebuttal: Question**
>
> >Q1: Could the model-based estimation in PALACE be integrated with CoIn’s verification to combine predictive power with cryptographic reliability?
>
> Yes, PALACE can be **naturally integrated** with hash-based verification methods such as CoIn. The two approaches address different components of the auditing pipeline and are therefore fully compatible. PALACE provides predictive auditing of hidden reasoning tokens, which cannot be cryptographically verified because providers do not expose internal traces. Hash-based verification methods such as CoIn check the integrity of visible tokens through cryptographic commitments. Using both together can improve the robustness of the auditing pipeline and increase the cost of inflation from COLS.
>
> >Q2: Have you tested whether PALACE’s GRPO-based length predictor generalizes across model families trained with different reasoning styles?
>
> Yes. We tested cross-model transfer by using a predictor trained on DeepSeek-R1 traces to estimate ChatGPT’s hidden reasoning tokens, and it completely failed to transfer (see the table we listed below). This shows that reasoning-length behavior is strongly model-specific, so auditing cannot rely on cross-family data; users cannot audit COLS using only open-source traces, and COLS themselves also cannot self-validate their billing via transfer. This is exactly why **minimal model-specific auxiliary auditing data are necessary and why PALACE advocates for releasing such datasets**.
>
> | Model | PALACE Pass@1 (General) | GPT o-3 Pass@1 (General) |
> |--------------|--------------------------|--------------------------|
>  | Qwen2.5-3B | 87.28 | 23.40 |
> | Qwen2.5-1.5B | 88.40 | 22.88 |
> | LLaMA3.2-3B | 81.42 | 34.01 |
>  | LLaMA3.2-1B | 79.42 | 34.95 |
>
> GPT o-3 Pass@1 values are measured by applying the same model trained on dataset from **DeepSeek-R1-Distill-Llama-70B** and treating GPT o-3 reported token counts as ground truth.
>
> We appreciate the reviewer’s constructive insights, and we believe the clarifications above further strengthen the contribution and practicality of PALACE as a first step toward standardized auditing of hidden reasoning tokens.

---

### Author Response · Authors · 2025-11-25
**General Response**

We thank all reviewers for their thoughtful and detailed feedback. Several reviewers highlighted the question of how much PALACE depends on provider-released calibration data and whether predictive auditing can generalize across domains or models. We summarize our position here.

**1. Predictive auditing is inherently model-family–specific.**
Our experiments show that hidden-reasoning behavior does not transfer across model families. A predictor trained on one family performs poorly on another. This is a limitation of the auditing problem itself, not of PALACE: different LLM families simply have different internal reasoning styles.

**2. Within each model family, generalization is strong and requires only a few adapters.**
Our cross-domain analysis demonstrates that a single general adapter already transfers well across diverse reasoning tasks. Domain-specific adapters (math, coding, medical) are only needed because modern COLS apply strong domain-specialized optimizations in these areas. In practice, for the few major model families, only 4–5 lightweight GRPO adapters per family are sufficient to cover multi-task auditing. This makes the data requirement modest and feasible.

**3. The provider-data question reflects the current ecosystem, not a flaw in PALACE.**
At present, no COLS releases verified reasoning-trace samples, which makes all forms of hidden-token auditing impossible. Our work does not imply that any provider is inflating usage, but highlights that inflation could occur and users currently have no way to verify it. If users and the research community recognize the importance of transparency, COLS will have strong incentives to release small, standardized calibration sets, which is similar to the evaluation benchmarks they already published.

**4. PALACE is the first predictive auditing framework and aims to define the problem clearly.**
Our main contribution is to articulate the hidden-reasoning auditing challenge and demonstrate, through experiments, that predictive auditing is both feasible and effective once minimal calibration data exist. We hope the significance of this direction and the value of establishing the first predictive auditing framework can be taken into account in the final scores.

We thank the reviewers again for their constructive input and hope that the significance of the problem and the novelty of our approach can be taken into account in the final evaluation.

---

### Comment · Area_Chair_qPQq · 2025-11-28

Dear Reviewers,

Thank you for your time and effort in serving as a reviewer.

The authors have now submitted their rebuttal, and this AC kindly asks you to review their response and assess whether your comments have been adequately addressed. If any points require further clarification or discussion, please feel free to raise those questions by adding comments and to initiate discussion as needed.

ICLR encourages reviewers to actively engage in the discussion phase, so your prompt actions are especially valuable. Thank you very much for your continued efforts and valuable contributions.

Best regards,

Your AC

---

### Meta-Review · Area_Chair_HYi9 · 2026-01-17

**Summary:**

This paper introduces PALACE, a user-side auditing framework for estimating hidden reasoning token usage in commercial LLMs. PALACE trains a token-count prediction model via supervised fine-tuning (SFT), further augmented with GRPO to learn lightweight, domain-specific adapters. The authors show that PALACE outperforms several baselines across multiple domains.

**Reviewer Concerns:**

Several reviewers raise substantial concerns regarding the generalizability and real-world practicality of PALACE (wSQr, e5tK, mR1Q, RsSN). A fundamental limitation is that the approach relies on provider-supplied training data to perform SFT and calibration, which are typically unavailable or unverifiable in realistic auditing scenarios. This requirement undermines the applicability of PALACE in fully black-box and cross-provider settings, which are central to the motivating use cases of auditing commercial LLM APIs.
Reviewers also express concerns about the use of a fixed 33% relative error threshold as the primary success criterion (e5tK, RsSN). This threshold lacks theoretical justification and is not supported by broad empirical validation across models, and domains.
Additional concerns include cross-domain performance and computational overhead. While the rebuttal addresses some points with additional experiments and clarifications, these responses do not fully resolve the core issues related to data assumptions, error tolerance, and practical deployment, which weaken confidence in the real-world applicability of PALACE.

**Reviewer Scores:**

Reviewers wSQr, e5tK, mR1Q, and RsSN primarily raised concerns about the real-world practicality of PALACE, which are not fully resolved by the rebuttal. In particular, the reliance on provider-supplied training and calibration data, as well as limited cross-model generalization, remains a potential obstacle to practicality. As a result, these reviewers would be likely to maintain their original scores.
Reviewer 6xRC raised additional concerns regarding the lack of theoretical analysis in PALACE, a point also echoed by reviewers e5tK and RsSN, and this concern was not addressed during the rebuttal. As a result, Reviewer 6xRC would also be likely to maintain the original score.

---

### Decision · Program_Chairs · 2026-01-26

Reject